# Intelligent Identification of Tea Plant Seedlings Under High-Temperature Conditions via YOLOv11-MEIP Model Based on Chlorophyll Fluorescence Imaging

**DOI:** 10.3390/plants14131965

**Published:** 2025-06-27

**Authors:** Chun Wang, Zejun Wang, Lijiao Chen, Weihao Liu, Xinghua Wang, Zhiyong Cao, Jinyan Zhao, Man Zou, Hongxu Li, Wenxia Yuan, Baijuan Wang

**Affiliations:** 1College of Mechanical and Electrical Engineering, Yunnan Agricultural University, Kunming 650201, China; chunwangkk@163.com; 2Yunnan Organic Tea Industry Intelligent Engineering Research Center, Kunming 650201, China; wangzejun0529741x@163.com (Z.W.); 2015056@ynau.edu.cn (L.C.); liuweihao2210@163.com (W.L.); wangaoyu2011@163.com (X.W.); czy@ynau.edu.cn (Z.C.); zhao29502@163.com (J.Z.); 15770383235@163.com (M.Z.); 13085315910@163.com (H.L.); 3College of Tea, Yunnan Agricultural University, Kunming 650201, China; 4Fengqing County Farmer Yuanshi Technology Service Station, Lincang 675900, China

**Keywords:** chlorophyll fluorescence imaging, high-temperature stress, tea plant seedlings, improved YOLOv11, model lightweight, image classification recognition

## Abstract

To achieve an efficient, non-destructive, and intelligent identification of tea plant seedlings under high-temperature stress, this study proposes an improved YOLOv11 model based on chlorophyll fluorescence imaging technology for intelligent identification. Using tea plant seedlings under varying degrees of high temperature as the research objects, raw fluorescence images were acquired through a chlorophyll fluorescence image acquisition device. The fluorescence parameters obtained by Spearman correlation analysis were found to be the maximum photochemical efficiency (Fv/Fm), and the fluorescence image of this parameter is used to construct the dataset. The YOLOv11 model was improved in the following ways. First, to reduce the number of network parameters and maintain a low computational cost, the lightweight MobileNetV4 network was introduced into the YOLOv11 model as a new backbone network. Second, to achieve efficient feature upsampling, enhance the efficiency and accuracy of feature extraction, and reduce computational redundancy and memory access volume, the EUCB (Efficient Up Convolution Block), iRMB (Inverted Residual Mobile Block), and PConv (Partial Convolution) modules were introduced into the YOLOv11 model. The research results show that the improved YOLOv11-MEIP model has the best performance, with precision, recall, and mAP50 reaching 99.25%, 99.19%, and 99.46%, respectively. Compared with the YOLOv11 model, the improved YOLOv11-MEIP model achieved increases of 4.05%, 7.86%, and 3.42% in precision, recall, and mAP50, respectively. Additionally, the number of model parameters was reduced by 29.45%. This study provides a new intelligent method for the classification of high-temperature stress levels of tea seedlings, as well as state detection and identification, and provides new theoretical support and technical reference for the monitoring and prevention of tea plants and other crops in tea gardens under high temperatures.

## 1. Introduction

The tea plant (*Camellia sinensis* (L.) Kuntze), a perennial evergreen economic crop belonging to the Theaceae family and the Camellia genus, prefers to grow in warm and humid climate environments [1]. As a thermophilic, hygrophilic, and photophobic economic crop, its optimal growth temperature range is between 20 and 25 °C, with a tolerance to high temperatures ranging from 35 to 40 °C. However, when the temperature rises above 40 °C, the tea plant will suffer severe damage, and its critical survival temperature is 45 °C [2,3]. In recent years, with global climate change, global warming has become increasingly evident, and extreme weather events have occurred more frequently. This has made tea plants more susceptible to the ravages of high temperature [4], drought [5], frost [6], and other extreme environmental factors. These adverse factors have had a significant negative impact on the growth and development of tea plants, not only leading to a decline in tea quality but also causing tea plant death in severe cases, thus posing a serious challenge to the sustainable development of the tea industry.

China is one of the primary birthplaces of tea trees. Tea, a cash crop, not only embodies the characteristics of Chinese agriculture but also serves as a significant vehicle for cultural heritage, carrying substantial social and economic value [7]. The tea-growing regions in China are primarily located in subtropical and tropical zones. Affected by the subtropical high-pressure system, the main tea-producing areas in China, such as the Jiangnan tea area, South China tea area, and Southwest China tea area [8], frequently experience prolonged high-temperature drought weather in summer, posing a serious threat to tea gardens, especially young tea gardens, and having a significant negative impact on the quality and yield of tea, leading to huge economic losses. Extremely high temperatures will lead to the emergence of new tea-reduced production areas. It is estimated that due to extreme high-temperature weather, the tea yield losses in the Yangtze River coastal areas and South China regions of China will reach 14–26%, while yield losses in provinces such as Chongqing, Hunan, Anhui, and Zhejiang are expected to be as high as 11–24% [9]. Yunnan province, as one of the important tea-producing areas in China, is famous for its rich variety of tea plants and excellent tea quality [10]. However, in recent years, extremely high-temperature weather has frequently affected tea production and quality to varying degrees, which poses an undeniable challenge to the development of the local tea industry. Against this backdrop, it is particularly important to delve into the response patterns of tea plants to high-temperature stress. This not only helps to implement more effective tea garden management measures but also provides timely and effective prevention and control strategies for possible extreme-high-temperature events in the future, thereby ensuring the sustainable development of the tea industry.

Photosynthesis not only sustains the rich and diverse ecosystems and biodiversity on Earth but also plays a critical role in maintaining the planet’s ecological balance, as well as the stability and survival of the biosphere [11]. However, high-temperature stress can affect the normal physiological and metabolic processes of plants at multiple levels, thereby inhibiting plant growth and development [12]. Notably, photosynthesis is one of the most temperature-sensitive physiological processes in plants. With the rapid development of chlorophyll fluorescence measurement and analysis techniques, an increasing number of researchers are engaged in the study of the photoinhibition of photosynthesis. Chlorophyll fluorescence imaging technology, as an emerging non-destructive detection method, has shown great potential in monitoring the physiological status of plants [13]. By utilizing chlorophyll fluorescence kinetics, it is possible to rapidly, sensitively, and non-invasively study and detect the impact of various stressors on plant photosynthetic physiology [14]. Furthermore, chlorophyll fluorescence parameters can, to a certain extent, reflect the impact of environmental factor changes on plant photosynthetic systems and their mechanisms of action. Jain, N. et al. found that high-temperature stress significantly affected the yield of wheat during the grain-filling stage, and chlorophyll fluorescence transient technology could effectively detect the damage to photosynthesis from high temperatures. The study used chlorophyll fluorescence parameters to screen heat-tolerant wheat genotypes in the field, and the results showed that specific genotypes maintained better photosynthetic efficiency under heat stress [15]. Guidi, L. conducted an in-depth exploration of the theoretical basis of chlorophyll fluorescence and its key parameters, as well as their important roles in assessing Photosystem II (PSII) photochemistry and energy dissipation. The research not only involved the relationship between actual PSII photochemistry and CO_2_ assimilation but also analyzed the roles of photochemical and non-photochemical quenching in the changes in PSII activity. Additionally, the application of chlorophyll fluorescence imaging technology in the study of leaf heterogeneity was discussed, and the impact of these parameters under different environmental stresses (such as water, nutrition, pollution, temperature, and salinity) was summarized [16].

In recent years, high-throughput phenotyping technology, which offers a rapid, non-destructive, and multi-dimensional analysis of plant traits, has significantly improved the efficiency of crop breeding and environmental response research. Muhammad Akbar Andi Arief et al. developed a non-destructive chlorophyll fluorescence imaging (CFI) system capable of effectively measuring the maximum photochemical efficiency (Fv/Fm) in strawberry plants. This system can also capture the spatiotemporal dynamics of plant responses to abiotic stresses, such as high temperature and drought, with great accuracy [17]. Mangalraj Poobalasubramanian et al. employed hyperspectral images to extract chlorophyll fluorescence indices. By analyzing these indices, they developed machine learning models to identify early high-temperature stress and water stress in strawberry plants. Their study found that the random forest classifier achieved an accuracy of 94%, while the gradient boosting classifier achieved 91% accuracy [18]. Chen et al. utilized chlorophyll fluorescence imaging technology to perform an early diagnosis of downy mildew in greenhouse cucumber seedlings. They screened chlorophyll fluorescence parameters using machine learning methods such as LASSO regression and then used a convolutional neural network method with transfer learning based on an improved ResNet50 structure to detect downy mildew in cucumber fluorescence images, achieving an accuracy of 94.76% [19]. Long et al. collected chlorophyll fluorescence parameters and corresponding images of tomato seedlings under four different drought stress levels. They analyzed drought stress categories through histogram characteristics and texture features, calculated the Pearson correlation of feature parameters, and input the highly correlated features into three models, Linear Discriminant Analysis (LDA), Support Vector Machine (SVM), and k-Nearest Neighbor (KNN), to determine the drought stress category. The recognition accuracies of the LDA, SVM, and KNN models were 86.8%, 87.1%, and 76.5%, respectively [20]. Concurrently, with the rapid advancement of deep learning technology, object detection algorithms have been widely applied across multiple fields. The YOLO series of algorithms, as classic models in the realm of real-time object detection, has garnered extensive attention due to their swift detection speeds and high levels of detection accuracy. YOLOv11, the latest iteration of this series, has achieved a significant enhancement in both precision and speed compared to its predecessors. It employs an improved version of CSPDarknet53 as its backbone network and introduces the Spatial Pyramid Pooling Fast (SPFF) module and the C2PSA module with a pyramid slicing attention mechanism, thereby further bolstering its feature extraction capabilities. Additionally, the detection head of YOLOv11 adopts a decoupled structure, selecting appropriate loss functions based on different tasks, which markedly improves the model’s performance. These refinements enable YOLOv11 to excel in processing large-scale image data, thereby furnishing a robust technological foundation for our research.

To date, research on heat stress in tea plants has predominantly focused on the measurement of basic photosynthesis-related indicators, while studies on the mechanisms of heat stress damage to tea plants are relatively scarce, especially in the light reactions of photosynthesis, where relevant research is rarely conducted. This study takes the “Yunkang 10” tea seedlings as the research object. By analyzing the fluorescence parameters and images of tea seedling leaves under different high-temperature stresses obtained through chlorophyll fluorescence imaging technology, and in combination with the proposed improved YOLOv11 model, it achieves efficient, non-destructive, and intelligent identification of the status of tea seedlings under various high-temperature stresses. This study aims to explore the potential application of chlorophyll fluorescence imaging and deep learning technology in the stress monitoring of tea seedlings, providing a rapid and accurate monitoring method for the high-temperature stress status of tea seedlings. The following are the main research contributions of this paper: (1) We systematically analyzed the effects of heat stress on the photosynthetic light reactions of tea plants. We comprehensively evaluated 13 chlorophyll fluorescence parameters, and through Spearman correlation analysis, the key indicator with the highest correlation to heat stress levels was determined as the maximum photochemical efficiency (Fv/Fm). Furthermore, using chlorophyll fluorescence imaging analysis, we delved into the impact of heat stress on the chlorophyll fluorescence characteristics of tea plant seedlings. (2) In order to reduce the number of model network parameters, enhance the efficiency and accuracy of feature extraction, decrease computational redundancy and memory access volume, and achieve higher model precision and efficiency while maintaining a low computational cost, the MobileNetV4 network and EUCB (Efficient Up Convolution Block), iRMB (Inverted Residual Mobile Block), and PConv (Partial Convolution) modules were introduced into the YOLOv11 model, respectively. Compared with YOLOv11, YOLOv10, YOLOv8, SSD, Faster-RCNN, and other models, the efficiency and accuracy of high-temperature stress-level classification and recognition were significantly improved, providing a promotable technical solution for intelligent phenotypic analysis in agriculture. (3) By combining chlorophyll fluorescence imaging technology with the lightweight YOLOv11-MEIP model, a tea plant stress status recognition system was established, achieving non-destructive and high-precision dynamic monitoring. This system provides a novel technological approach for investigating plant stress physiology.

## 2. Materials and Methods

### 2.1. Experimental Samples

In this study, two-year-old Yunnan large-leaf tea seedlings of the “Yunkang No. 10” variety were used as experimental subjects. The experiment was conducted at the Tea College of Yunnan Agricultural University from September to November 2024. The tea seedlings were planted in plastic pots with a diameter of 16 cm and a height of 17 cm and were cultivated in a constant temperature environment of 20 °C for 14 days to ensure that the tea seedlings adapted to the new growing environment. A total of 30 healthy seedlings, characterized by similar stem thicknesses and good growth statuses, were selected and assigned to experimental and control groups. Using a temperature- and humidity-controlled incubator, the seedlings were subjected to high-temperature stress at four temperature levels: 25 °C (control), 30 °C, 36 °C, and 42 °C. The control group was maintained at 25 °C for 3 days, while the experimental group seedlings were exposed to high-temperature stress at 30 °C, 36 °C, and 42 °C for 3 days. The model of the temperature and humidity incubator used was RQX-300H (Shanghai Yuejin Medical Instrument Co., Ltd., Shanghai, China), with a temperature control range of 5~50 °C, a humidity control range of 50~90%RH, a light intensity of 12,000 Lx, a temperature fluctuation of ±1.5 °C, and a temperature uniformity of ±2 °C. The environmental settings are shown in Table 1. According to the different temperature levels set in the incubator, the high-temperature stress of tea seedlings was divided into four levels: CK (control), LV1, LV2, and LV3. White-light illumination at 200 $\mu molm^{-2}s^{-1}$ was provided by specialized high-brightness fluorescent lamps in the culture chamber to simulate natural daytime conditions, while turning off the lamps during these periods simulated nighttime conditions.

### 2.2. Fluorescence Image Acquisition Equipment

In this study, the PlantExplorer series multifunctional plant photosynthetic phenotyping measurement system (PhenoVation B.V., Wageningen, The Netherlands) was used to acquire chlorophyll fluorescence parameters and fluorescence images of tea plant seedlings, as shown in Figure 1. This system employs innovative multispectral chlorophyll fluorescence and visible light imaging technology, combined with the latest LED technology, CCD technology, and communication technology, enabling precise measurements of plant phenotypes. It is capable of acquiring chlorophyll fluorescence imaging, in addition to RGB, chlorophyll, and anthocyanin imaging, with an imaging area of 40 cm × 53 cm. The device is equipped with a built-in four-channel multispectral LED system as its photosynthetic light source, covering two channels of white light, red light, blue light, and far-red light. Precise spectral control is enabled by this system, which is useful for studying plant responses to varying light wavelengths. At 60 cm from the light source, the photosynthetic light intensity ranges from 100 to 600 $\mu molm^{-2}s^{-1}$ and is adjustable. To ensure that the reaction centers of PSII are in a fully open state, a dark treatment operation of approximately 20 min is required for the tea plant seedlings to be tested before the acquisition of chlorophyll fluorescence data. Following dark adaptation, fluorescence parameters and images of tea seedlings were measured using the instrument, with each sample measurement taking approximately 11 to 12 min.

### 2.3. Data Acquisition and Processing

#### 2.3.1. Correlation Analysis of Fluorescence Parameters

Chlorophyll fluorescence parameter values can be directly measured through the multifunctional plant photosynthetic phenotyping measurement system and are presented in a numerical visualization format. In addition to the values provided by the system, the measurement yields 13 distinct chlorophyll fluorescence parameter values. These parameter values can serve as a basis for determining whether tea plant seedlings are affected by high-temperature stress, and their specific meanings are shown in Table 2 [21,22]. The formulas for calculating photosynthetic parameters are presented in Equations (1)–(5), where F refers to the fluorescence intensity recorded under photoadaptive conditions prior to the application of a saturating pulse.(1)$$Fv/Fm=\frac{Fm-Fo}{Fm}$$(2)$$qP=\frac{Fm^{\prime }-F}{Fv^{\prime }}=1-\frac{F-Fo^{\prime }}{Fm^{\prime }-Fo^{\prime }}$$(3)$$qL=qP\times \frac{Fo^{\prime }}{F}$$(4)$$qN=\frac{Fv-Fv^{\prime }}{Fv}=1-\frac{Fm^{\prime }-Fo^{\prime }}{Fm-Fo}$$(5)$$NPQ=\frac{Fm-Fm^{\prime }}{Fm^{\prime }}$$

A Spearman correlation coefficient analysis [23] was conducted on the 13 chlorophyll fluorescence parameters of tea plant seedlings under high-temperature stress. When calculating the correlation coefficient of Spearman grade, the quantitative value of the high-temperature stress level is based on different high-temperature stress levels set in the experimental design, which is used for correlation analysis with chlorophyll fluorescence parameters to determine the parameters significantly related to high-temperature stress. With the results shown in Table 3, the analysis revealed that five characteristic parameters had high correlations with the stress levels, among which the fluorescence parameter Fv/Fm had the highest correlation with the stress levels, at −0.886. Therefore, this study selected the fluorescence parameter Fv/Fm as the key indicator for determining the high-temperature stress levels.

#### 2.3.2. Fluorescence Image Analysis

A grayscale histogram intuitively reflects the distribution of each grayscale level in a grayscale image and serves as an important tool for image difference analysis [24]. It reflects the frequency of occurrence of each grayscale level in the image by counting the number of pixels at each grayscale level. In the grayscale histogram, the horizontal axis represents the grayscale levels, and the vertical axis represents the frequency of occurrence of each grayscale level. As a function of grayscale levels, the grayscale histogram usually normalizes the vertical axis to the interval [0, 1] to more clearly display the distribution of each grayscale level, and its specific calculation formula is shown in Equation (6).(6)$$P\left(r_{k}\right)=\frac{n_{k}}{MN}$$

In the formula, $r_{k}$ represents the grayscale level of a pixel, $n_{k}$ is the number of pixels with grayscale level $r_{k}$, and M and N are the number of rows and columns in the image, respectively, i.e., the total number of pixels in the image.

Figure 2 displays the fluorescence images of the chlorophyll fluorescence parameter Fv/Fm, along with their corresponding grayscale histograms. According to the results shown in the figure, when the tea plant seedling leaves are in a healthy state, the grayscale value distribution range of the Fv/Fm parameter is [0.7, 0.9], and the fluorescence image appears green. Under high-temperature stress level LV1, the grayscale value distribution range changes to [0.5, 0.9], with some grayscale values rising to the range of [0.5, 0.7]. The edges of the leaves begin to turn yellow, and some leaves turn completely yellow. At high-temperature stress level LV2, the grayscale value distribution range further changes to [0.3, 0.9], with about one-third of the grayscale values located within the range of [0.3, 0.7]. Most of the leaves appear yellow, and certain areas begin to turn red. At stress level LV3, the grayscale value distribution range is [0.1, 0.9], with grayscale values transitioning from the range of [0.7, 0.9] to [0.1, 0.3], and some grayscale values rising to the range of [0.1, 0.3]. Most of the leaf edges are reddish-brown, with only a small part of the veins still maintaining a yellow-green color.

#### 2.3.3. Establishment of the Image Dataset

Chlorophyll fluorescence imaging technology is an advanced non-invasive detection method that holds a crucial position in the field of plant scientific research. In the study of plant stress responses, chlorophyll fluorescence imaging technology can sensitively capture the physiological responses of plants to environmental stresses such as drought, high temperature, and salinity [25], thereby offering vital information for research on stress physiology. As shown in Figure 3a, the comparison of RGB images and chlorophyll fluorescence parameter Fv/Fm images of tea seedlings across the CK, LV1, LV2, and LV3 temperature groups is displayed. Corresponding to the measured fluorescence parameters, the Fv/Fm parameter range for the CK temperature group is between 0.77 and 0.81. For the LV1 temperature group, the Fv/Fm parameter range is between 0.70 and 0.75. For the LV2 temperature group, the Fv/Fm parameter range is between 0.56 and 0.66. Lastly, in the LV3 temperature group, the Fv/Fm parameter range is between 0.18 and 0.60.

A total of 244 raw chlorophyll fluorescence images were collected. However, this number was insufficient for training a high-performance object detection model. To mitigate the risk of overfitting during the model training phase and enhance the model’s generalization ability, a series of image augmentation techniques was applied to the original experimental dataset. These techniques included horizontal flipping, rotation transformation, brightness adjustment, and changes in chroma, contrast, and sharpness, thereby significantly increasing the scale of the image dataset. The augmented fluorescence images are shown in Figure 3b. The expanded fluorescence image dataset consists of a total of 1952 images, which were randomly divided into training, validation, and testing sets in the ratio of 8:1:1. The training set comprises 1562 images, while the validation and testing sets each consist of 195 images.

### 2.4. Improvement of the YOLOv11 Network

YOLOv11, an object detection algorithm released by Ultralytics on 30 September 2024, represents a significant advancement in both accuracy and speed compared to previous models in the YOLO series. The YOLOv11 model adopts an improved version of CSPDarknet53 as its backbone network, replacing the original C2f module with the C3K2 module. In the backbone network, the CBS (convolution, batch normalization, and SiLU activation function) module first performs convolution, followed by batch normalization, and subsequently enhances the output through the SiLU activation function. Furthermore, the backbone network introduces the Spatial Pyramid Pooling Fast (SPFF) module, which pools feature maps to a fixed size, thereby increasing the diversity of feature representation. The feature extraction capability is further enhanced through the C2PSA module, which employs a pyramid slice attention mechanism. This PSA mechanism adopts a multi-level design that improves the SE attention mechanism, making it more adept at processing multi-level features. The neck network uses a PAN-FPN structure, which enhances the fusion of shallow location information and deep semantic information through a bottom-up path, supplementing the deficiency of object localization information in the FPN structure. The detection head of YOLOv11 adopts a decoupled structure, with independent branches for predicting class and location information, and appropriate loss functions are chosen for different tasks. The binary cross-entropy loss (BCELoss) is used for classification tasks, while the distribution focal loss (DFL) and CIoU are adopted for bounding box regression tasks. In addition, two depthwise convolution (DWConv) layers have been added to the classification detection head, significantly reducing the number of parameters and computational overhead [26,27].

In this study, an improved lightweight YOLOv11 network model was proposed to recognize the fluorescence images of tea plant seedlings under different high-temperature levels. The structure of the improved network model is shown in Figure 4: (1) To reduce the number of network parameters and improve model accuracy and efficiency while maintaining a low computational cost, thereby laying the foundation for subsequent deployment on mobile devices, the MobileNetV4 network was used to improve the YOLOv11 backbone network. (2) To achieve efficient feature upsampling and fusion, enhancing the model’s computational efficiency and feature expression capabilities, the Efficient Up Convolution Block (EUCB) module was introduced into the YOLOv11 network. (3) To maintain a lightweight model while significantly improving the efficiency and accuracy of feature extraction, the Inverted Residual Mobile Block (iRMB) module was introduced into YOLOv11. (4) The Partial Convolution (PConv) module was introduced into the model to reduce computational redundancy and memory access volume.

#### 2.4.1. Target Detection Network MobileNetV4

With the increasing complexity of visual tasks and the growing demand for real-time performance, traditional convolutional neural network architectures struggle to meet the needs for efficient deployment on mobile devices. Although YOLOv11 is a powerful model, it consumes substantial computational resources when processing large-scale image data, which limits its applicability on resource-constrained mobile platforms. To address this limitation, our study incorporates MobileNetV4 as the backbone network for YOLOv11, enabling the model to maintain efficient computation while achieving more accurate feature extraction. This improvement makes YOLOv11 more suitable for deployment on mobile devices.

MobileNetV4 [28] represents the latest iteration of the MobileNet series, featuring a universal and efficient architectural design. Its core innovations include the Universal Inverted Bottleneck (UIB) search block, Mobile Multi-Query Attention (Mobile MQA), and an optimized neural architecture search (NAS). The UIB integrates the MobileNet Inverted Bottleneck (IB) block [29], ConvNext [30], Feed-Forward Network (FFN) [31], and a novel Extra Depthwise (ExtraDW) variant. As shown in Figure 5, the UIB block contains two optional depthwise convolutions with four possible instantiations. The MobileNet Inverted Bottleneck performs spatial mixing on the expanded features to increase model capacity at the cost of higher computation. ConvNext allows for larger kernel sizes in spatial mixing at a lower cost by performing spatial mixing before expansion. ExtraDW combines the advantages of ConvNext and IB, increasing network depth and receptive field while minimizing computational cost. FFN comprises two stacked 1×1 pointwise convolutions with activation and normalization layers in between. This design allows for the expansion of the receptive field and enhancement of feature extraction capabilities at each network stage, maximizing computational utilization.

To optimize the performance of mobile accelerators, MobileNetV4 introduces an attention module called Mobile MQA. This module is optimized for the computational and storage limitations of mobile devices and can provide up to a 39% increase in inference speed [28]. Mobile MQA further integrates Spatial Reduction Attention (SRA), which reduces the resolution of keys and values through asymmetric spatial downsampling operations while maintaining the high resolution of queries. In addition, the module leverages the correlation between spatially adjacent tokens in hybrid models, replacing the average pooling operation with a 3×3 depthwise convolution with a stride of 2 to achieve spatial reduction. The principle is shown in Equations (7) and (8), where X is the input feature map, $W^{O}$ is the output weight matrix, $W^{Q_{j}}$ is the weight matrix of the j-th query, $W^{K}$ is the weight matrix of the key, $W^{V}$ is the weight matrix of the value, SR represents spatial reduction, i.e., depthwise convolution with a stride of 2, and $d_{k}$ is the dimension of the key.(7)$$Mobile\_MQA(X)=Concat({attention}_{1},\ldots ,{attention}_{n})W^{O}$$(8)$${attention}_{j}=softmax(\frac{{\left(XW^{Q_{j}}\right)\left(SR\left(X\right)W^{K}\right)}^{T}}{\sqrt{d_{k}}})(SR(X)W^{V})$$

MobileNetV4 adopts an optimized neural architecture search (NAS) method, which significantly enhances the effectiveness of model search. This approach implements efficient model architecture optimization through a two-stage search strategy. In the initial coarse-grained search phase, the focus is on identifying the optimal filter size while keeping other parameters fixed. Subsequently, during the fine-grained search phase, the configuration of the two depth layers in the UIB is further optimized, including the selection of kernel sizes and their presence. Moreover, this NAS method integrates an offline distillation dataset to reduce noise in NAS measurements, thereby improving the overall model quality.

#### 2.4.2. Efficient up Convolution Block

In the field of deep learning, upsampling and convolution operations are crucial steps in tasks such as image reconstruction, super-resolution, and semantic segmentation. However, traditional upsampling methods often suffer from low computational efficiency, a large number of parameters, and insufficient feature fusion. To address these issues, we introduce an efficient upsampling module, the Efficient Up Convolution Block (EUCB) [32], aiming to achieve efficient feature upsampling and fusion through innovative structural design and optimization strategies, thereby enhancing the model’s computational efficiency and feature expression capability. The structure of the EUCB is shown in Figure 6.

The EUCB upsamples the feature map of the current stage step by step to match its dimensions and resolution with those of the feature map in the next skip connection. First, the EUCB performs an upsampling operation, using upsampling with a scale factor of 2 to double the size of the input feature map. Subsequently, a 3×3 DWConv [33] is applied, followed by batch normalization (BN) and ReLU activation. These steps efficiently enhance the feature map without significantly increasing the computational load. Finally, a 1×1 convolution is used to reduce the number of channels, matching the upsampled feature map with the number of channels in the next stage. In the EUCB, depthwise convolution is used instead of standard 3×3 convolution to achieve efficient computation, making it particularly suitable for applications with high computational efficiency requirements. The calculations for the EUCB are shown in Equations (9)–(11).(9)$$EUCB(x)=C_{1\times 1}(ReLU(BN(DWConv(Up(x)))))$$(10)$$ReLU\left(x\right)=\left\{\begin{array}{c} x,\;x>0\\ 0,\;x\leq 0 \end{array}\right.$$(11)$$BN=y_{i}=\gamma \frac{x_{i}-\mu }{\sqrt{\sigma^{2}+\varepsilon }}+\beta$$

#### 2.4.3. Inverted Residual Mobile Block

In today’s computer vision field, with the continuous enhancement of mobile device computing power and the increasing complexity of visual tasks, the demand for efficient and lightweight neural network modules is also growing. Despite being an advanced real-time object detection model, YOLOv11 still faces dual challenges in feature extraction efficiency and accuracy. To address this issue, we introduced the iRMB (Inverted Residual Mobile Block) module into YOLOv11. The iRMB module combines the advantages of inverted residual structures and attention mechanisms, significantly improving feature extraction efficiency and accuracy while maintaining a lightweight model [29,34,35].

As shown in Figure 7, the design of iRMB integrates the efficiency of CNN-like operations for modeling local features and the dynamic modeling capability of Transformer-like mechanisms for learning long-range interactions. In the iRMB, F is constructed as a combination of serially connected Expanded Window Multi-Head Self-Attention (EWMHSA) and DWConv operations. In the EWMHSA, the parameters for obtaining Q and K are both based on the unexpanded image input X, which allows for more efficient computation of the attention matrix. Meanwhile, the expanded value $X_{e}$ is used as the value for parameter V, with the calculation principles shown in Equations (12)–(14). Through this connection method, the expansion speed of the model’s receptive field can be accelerated, resulting in a maximum path length of only L for the model, as shown in Equation (15).(12)$$F\left(\cdot \right)=(DWConv,Skip)(EWMHSA(\cdot ))$$(13)$$Q=K=X(\in R^{C\times H\times W})$$(14)$$V=X_{e}(\in R^{\lambda C\times H\times W})$$(15)L=O(2W/(k−1+2w))

#### 2.4.4. Lightweight Partial Convolution

After the model underwent lightweight processing, its overall floating-point operation count (FLOPS) was significantly reduced. However, the floating-point operations per second (FLOPs) did not show a significant improvement. Therefore, in order to achieve a dynamic balance between operator processing speed and the overall FLOPS of the model, we introduced Partial Convolution (PConv) to reduce computational redundancy and memory access volume [36,37].

The operating principle of PConv is illustrated in Figure 8. PConv applies regular convolution to only a portion of the input channels for spatial feature extraction, while keeping the remaining channels unchanged. As a result, the FLOPs of PConv are lower than those of regular Conv, and it has higher processing efficiency compared to DWConv and GConv [38]. For consecutive or regular memory access, the initial or terminal consecutive $C_{p}$ channels are used as representatives for computing the entire feature map. Under the premise of maintaining theoretical universality, assuming that the input and output feature maps have the same number of channels, the FLOPs of PConv are given by(16)$$h\times w\times k^{2}\times C_{P}^{2}$$
where h represents the column dimension, w represents the row dimension, $C_{p}$ is the feature dimension of PConv, and k is the convolutional kernel size. Under the typical partial ratio of $r=\frac{C_{p}}{C}=\frac{1}{4}$, the FLOPs of PConv are only $\frac{1}{16}$ of that of regular Conv. Meanwhile, the memory access volume of PConv is also relatively small, approximately(17)$$h\times w\times 2C_{p}+k^{2}\times C_{p}^{2}\approx h\times w\times 2C_{p}$$

At this point, the memory access volume of PConv with r=1/4 is only 1/4 of that of regular convolution. This indicates that, compared to regular convolution, PConv significantly reduces both floating-point operations and memory access volume. Consequently, this accelerates the model’s processing and improves its FLOPs. For the entire YOLOv11 model, replacing Conv with PConv optimizes the FLOPs for low-latency data processing. Further optimization and streamlining of the model make the batch-processed data more lightweight.

### 2.5. Evaluation Index and Operating Environment

To evaluate the overall performance of the improved YOLOv11 model, this study selected precision (P), recall (R), F1 score, Average Precision (AP), and mean Average Precision (mAP) as the evaluation metrics, which were used to assess the model’s classification and object detection performance [39,40]. The calculation expressions for these metrics are as follows:(18)$$Precision=\frac{TP}{TP+FP}$$(19)$$Recall=\frac{TP}{TP+FN}$$(20)$$F_{1}=2\times \frac{Precision\times Recall}{Precision+Recall}$$(21)$$AP=\int_{0}^{1}Precision(Recall)dRecall$$(22)$$mAP=\frac{1}{N}\sum_{i=1}^{N}AP(i)$$

In these formulas, TP represents the number of samples correctly identified by the model, FP represents the number of samples that actually exist but are incorrectly identified by the model, and FN represents the number of samples not detected by the model. To verify the effectiveness of the algorithm proposed in this paper and to ensure the rigor of the experimental training of each model, the same experimental platform and software version were used, as shown in Table 4.

## 3. Results

### 3.1. Ablation Study

In this study, we improved the YOLOv11 network model. To verify the effectiveness of the improved model, we conducted a statistical analysis of the effects of each module improvement, with the results shown in Table 5.

As shown in Table 5, after using the MobileNetV4 network as the backbone network of the YOLOv11 model, the number of model parameters was significantly reduced, while the model’s accuracy and efficiency were improved. The model’s precision and mAP50 increased by 0.9% and 1.17%, respectively. The performance indicators of parameters, gradients, GFLOPs, and weight were reduced by 16.2%, 16.2%, 1.2, and 1.6, respectively. With the improvement of the EUCB, the model’s upsampling process was optimized, thereby enhancing the model’s efficiency and feature expression ability when processing images. This resulted in an increase in precision and mAP50 by 0.94% and 1.5%, respectively. The optimization of the iRMB, through refining the residual block structure, further enhanced the model’s learning and generalization capabilities. This improvement led to increases in precision, recall, and mAP50 by 1.94%, 6.57%, and 2.87%, respectively, without significantly increasing the model’s complexity and computational burden, effectively boosting the model’s performance in visual tasks. The PConv improvement, which incorporated a lightweight structure, reduced the model’s parameters and computational load. The parameter and gradient indicators were both reduced by 3.2% compared to the original model, while GFLOPs and weight were reduced by 0.6 and 0.4, respectively. At the same time, the precision, recall, and mAP50 indicators were improved by 0.93%, 6.26%, and 2.27%, respectively.

Compared to the original YOLOv11 model, the YOLOv11-MEIP model achieved significant performance improvements. Although the number of layers in the improved model increased, the precision, recall, and mAP50 metrics were greatly enhanced, increasing by 4.05%, 7.86%, and 3.42%, respectively. The model’s parameters, gradients, and GFLOPs all decreased significantly by 29.45%, 29.45%, and 30.76%, respectively. The model’s weight also decreased by 1.6MB, further confirming the effectiveness of these module improvements on model performance. These enhancements not only improve the model’s detection accuracy but also increase the model’s applicability and reliability in future practical applications.

### 3.2. Loss Function Analysis

In the field of object detection, the loss function quantifies the difference between the model’s predicted outputs and actual results and serves as a key indicator for evaluating model performance. A decrease in the loss function value indicates an improvement in the model’s prediction accuracy, thereby reflecting an enhancement in the overall model performance. The loss function not only clarifies the optimization objective during the model training process but also significantly affects the model’s computational complexity and convergence speed. An optimal loss function can effectively guide the model to learn data features more precisely, thereby significantly improving the accuracy and efficiency of object detection [41].

As shown in Figure 9, the YOLOv11-MEIP model showed a significant convergence advantage in the early stages of training, with a faster loss function convergence speed than the YOLOv11 model. This indicates that the improved model can adapt to the training data more quickly. Notably, when the training iteration reached approximately 80 epochs, the convergence speed of the model’s loss function slowed significantly, which may suggest that the model’s parameter space is approaching its local or global optimum. Compared to the convergence slowdown of YOLOv11 at around 130 epochs, YOLOv11-MEIP entered the stable stage about 50 epochs earlier, which may imply that the improved model structure helps to accelerate the convergence process. As the number of training iterations continued to increase, the convergence of the loss function gradually stabilized. The boundary box loss, classification loss, and feature point loss of YOLOv11-MEIP eventually stabilized at around 0.59, 0.34, and 1.08, respectively, highlighting the excellent convergence stability of the improved model on the training set.

In the validation set, the YOLOv11 model exhibited significant fluctuations during the initial training phase. This unstable convergence could be attributed to the model’s insufficient adaptability to the distribution of the validation data. It was not until the training iteration reached approximately 250 epochs that the model achieved smooth convergence, and the fluctuation phenomenon was improved. In contrast, the training process of the YOLOv11-MEIP model was more stable, with the smoothness of the loss function training curve far exceeding that of the YOLOv11 model. This indicates that the improved model has better generalization ability on the validation set. The convergence speed of the loss function of the YOLOv11-MEIP model began to slow down after 50 training epochs, about 50 epochs earlier than the original YOLOv11 model, further confirming the rapid convergence characteristics of the improved model. After 500 epochs of training, the loss function convergence of the YOLOv11-MEIP model on the validation set tended to stabilize, with the bounding box loss, classification loss, and feature point loss stabilizing below 0.52, 0.18, and 0.96, respectively. This demonstrates the significant advantages of the improved model in reducing overfitting and enhancing generalization ability.

### 3.3. Comparative Experiments of Different Models

To further verify the advantages of the YOLOv11-MEIP model in detection performance, this study conducted experimental comparisons between the YOLOv11-MEIP model and other mainstream object detection models such as YOLOv10, YOLOv8, YOLOv7, YOLOv5, SSD, and Faster-RCNN. The results are shown in Table 6.

Based on the data presented in the table above, the YOLOv11-MEIP model proposed in this study exhibits outstanding performance in detecting chlorophyll fluorescence images of tea plant seedlings under various high-temperature conditions. The precision, recall, and mAP50 values reached by the model were 99.25%, 99.19%, and 96.46%, respectively. In comparison with other models in the YOLO series (v11, v10, v8, v7, and v5), precision increased by 4.05%, 4.76%, 6.11%, 7.89%, and 9.07%, respectively. Recall increased by 7.86%, 7.71%, 8.6%, 10.9%, and 8.4%, respectively. Meanwhile, mAP50 increased by 3.42%, 2.26%, 6.6%, 8.29%, and 5.56%, respectively. When compared to the single-object detection SSD model, the YOLOv11-MEIP model outperformed it with increases of 5.54%, 6.35%, and 4.94% in precision, recall, and mAP50, respectively. Compared with the two-stage object detection algorithm Faster-RCNN, the P, R, and mAP50 were slightly improved, with increases of 6.48%, 5.57%, and 4.23%, respectively. Additionally, the improved YOLOv11 model significantly reduced the number of GFLOPs, model parameters, and model weights, resulting in a substantial decrease in the resource requirements for model execution. This reduction in computational complexity provides a solid technical foundation for future applications of the model in resource-constrained environments.

This study also tested the detection performance of different models using the test set image data, with the results shown in Figure 10. The figure displays the recognition and detection results of the fluorescent images of tea plant seedlings under four temperature levels by different models, both with and without background removal. The results show that all eight models can effectively detect the target in the fluorescent images of tea plant seedlings. However, by examining the confidence of the model’s object detection, it can be observed that the YOLOv11-MEIP model has the highest confidence for target detection under different high-temperature conditions, followed by the YOLOv11 model, while the Faster-RCNN model has a relatively lower confidence. This indicates that the YOLOv11-MEIP model can better address the problem of target localization bias and avoid repeated detection. From the figure, it can be observed that when the YOLOv11-MEIP model successfully detects the target, its confidence is the highest among all models. In comparison, although the YOLOv11, YOLOv10, YOLOv8, YOLOv7, YOLOv5, SSD, and Faster-RCNN models can also achieve detection, their confidence levels slightly decrease, and some models exhibit issues of missed and false detections. In conclusion, from the perspective of confidence, the YOLOv11-MEIP model in this study shows the highest confidence, indicating its superior performance in object detection, as well as good robustness and generalization ability.

## 4. Discussion

By integrating chlorophyll fluorescence imaging technology with the improved YOLOv11-MEIP model, this study has realized efficient, non-invasive, and intelligent identification of the status of young plants of the large-leaf tea variety from Yunnan under various high-temperature stresses. The outcomes of this research not only offer novel technical approaches for monitoring high-temperature stress in tea plants but also provide significant technical support for further probing into the physiological response mechanisms of tea plants under high-temperature stress.

Firstly, chlorophyll fluorescence imaging technology demonstrated significant potential for monitoring plant physiological status. The studies by Jain, N. et al. [15], Guidi, L. et al. [16], and Muhammad Akbar Andi Arief et al. [17] have emphasized the effectiveness of chlorophyll fluorescence parameters in monitoring photosynthetic damage and assessing plant responses to stress by analyzing these parameters under various environmental stresses. In this study, the relevant fluorescence parameters and images of young plants of the large-leaf tea variety from Yunnan under different high-temperature stresses were analyzed, with the Fv/Fm parameter showing the highest correlation with the stress levels. Fv/Fm reflects the maximum potential efficiency of PSII and indicates the overall photosynthetic performance under stress. Therefore, the fluorescence images of Fv/Fm were selected as the input for the target detection model to accomplish the task of identifying the status of the seedlings.

Secondly, under different high-temperature stress conditions, the improved YOLOv11 model showed excellent performance in recognizing the fluorescent images of Yunnan large-leaf tea seedlings. By improving the YOLOv11 backbone network with the MobileNetV4 network, the model’s parameter size and computational demands are significantly reduced. This makes it more suitable for deployment on embedded resource-constrained mobile devices and enables on-site real-time monitoring. In addition, by introducing the EUCB, the model’s computational efficiency and feature expression ability were enhanced; the iRMB module was introduced to significantly improve the efficiency and accuracy of feature extraction; and the PConv module was introduced to reduce the model’s computational redundancy and memory access volume. Compared with the work of Chen [19] and Long [20], who effectively identified plant phenotypic states by combining chlorophyll fluorescence imaging with deep learning and machine learning models, the present study further extends the application of this technology to heat stress in tea plants. Through the improved YOLOv11-MEIP model, the high-precision identification of heat stress status in tea seedlings was achieved.

Despite the significant achievements in tea plant high-temperature stress recognition, this study still has limitations. Firstly, the study was primarily conducted in a controlled laboratory environment, which differs from the complex and variable conditions in actual fields. In practical applications, factors such as light intensity and angle, as well as background interference, may affect imaging quality and the model’s recognition performance. Secondly, the proposed model was trained based on specific tea plant seedlings and defined high-temperature stress conditions. In reality, tea plants exhibit a wide variety of species, which may differ in morphology, physiological characteristics, and response mechanisms to high temperatures. These differences could cause adaptability issues when applying the model to recognize high-temperature stress across different species, affecting the accuracy and reliability of recognition. In the future, we plan to expand the scope of our research by conducting experiments across multiple species and environments. We will perform high-temperature stress experiments on tea plant seedlings under actual field or potted conditions to verify and optimize our model. By enriching the diversity of tea plant species and integrating sample data from more regions, we aim to enhance the model’s adaptability and reliability in recognizing high-temperature stress in various tea species. Concurrently, future research endeavors will integrate the analysis of chlorophyll content to conduct a more comprehensive validation of the model’s predictive outcomes, thereby further refining the model and methodologies employed. This will improve the model’s applicability in real agricultural production and facilitate the translation of our research outcomes into practical applications for tea plant high-temperature stress monitoring and control. This will provide tea farmers and agricultural managers with more accurate and effective technical support.

By thoroughly analyzing the response of tea plant seedlings to high-temperature stress, this study can provide scientific guidance for the heat-resistant cultivation and management of tea plants. The improved YOLOv11-MEIP model, trained on chlorophyll fluorescence images labeled for heat stress, can effectively identify fluorescence feature changes caused by high temperatures. This integration of chlorophyll fluorescence imaging and deep learning offers a new method for monitoring heat stress in tea plants. It also provides key technical support and research directions for studying the physiological responses and heat-resistance mechanisms of tea plants under high-temperature conditions. Future research can further explore the differences in responses to high-temperature stress among different crops and how to better apply these technologies to agricultural production practices.

## 5. Conclusions

To enable efficient and accurate detection of the physiological characteristics of Yunnan large-leaf tea seedlings under varying high-temperature stress conditions, this study employed the multifunctional plant photosynthetic phenotyping measurement system, Plant Explorer Pro+, to measure the chlorophyll fluorescence parameters and capture images of the tea seedlings. Spearman correlation analyses were performed on these parameters, and it was found that five characteristic parameters had significant correlations with the stress levels, with Fv/Fm exhibiting the strongest correlation (a correlation coefficient of −0.886). Therefore, we selected Fv/Fm as the key feature parameter and used its corresponding fluorescence images as the input dataset for the deep learning model. This study improved the YOLOv11 model by replacing the original backbone network with the MobileNetV4 network. This modification significantly reduces the model’s parameter count and computational demands. Additionally, the upsampling module EUCB, the iRMB module, and PConv were incorporated to improve the model’s computational efficiency and feature representation capabilities. These enhancements notably increase the efficiency and accuracy of feature extraction while reducing computational redundancy and memory access volume. Compared with the original YOLOv11 model, the improved model reduced its parameter count by 29.4%, reduced its model weight by 1.6 MB (only 70.3% of the original), and decreased its GFLOPs from 6.5 to 4.5. At the same time, the model’s precision, recall, and mAP50 reached 99.25%, 99.19%, and 96.46%, respectively. When compared to other models in the YOLO series, as well as SSD and Faster-RCNN, the improved YOLOv11-MEIP model achieved a lightweight model while maintaining improved accuracy, demonstrating significant performance advantages.

This study integrates chlorophyll fluorescence imaging with an enhanced YOLOv11-MEIP model to precisely monitor the heat stress status of Yunnan large-leaf tea plant seedlings. This approach offers a novel perspective for investigating tea plants’ high-temperature stress response mechanisms. By correlating chlorophyll fluorescence images with model identification results, we can better understand the physiological changes and heat tolerance mechanisms in tea seedlings under heat stress. Our work provides a richer theoretical foundation and technical support for developing heat-resistant tea varieties and improving cultivation management practices. Future research will focus on more high-speed, convenient, and accurate high-temperature stress monitoring technologies based on the current foundation. The scale of the dataset will be expanded to show the high-temperature status of different types of tea seedlings. Further research on lightweight models will be carried out to achieve efficient deployment and application on edge devices.

## Figures and Tables

**Figure 1 plants-14-01965-f001:**
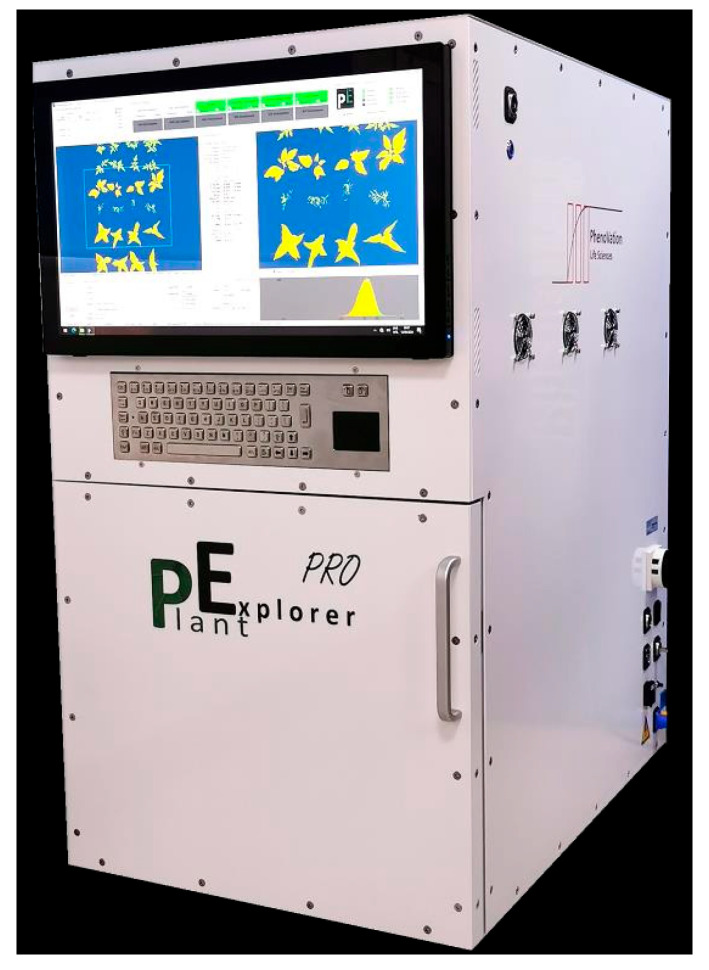
Multifunctional plant photosynthetic phenotyping measurement system.

**Figure 2 plants-14-01965-f002:**
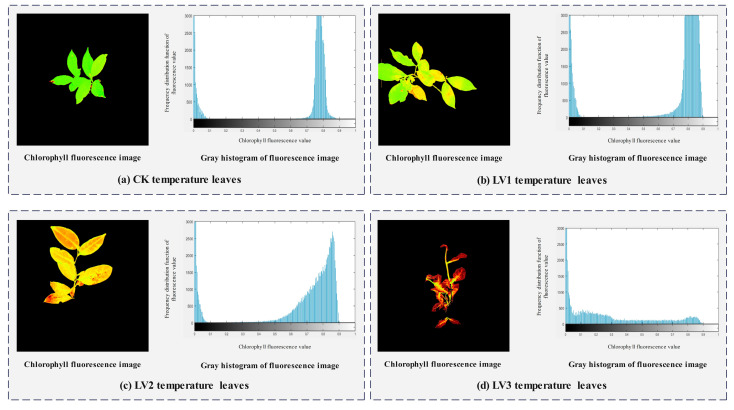
Chlorophyll fluorescence images and corresponding grayscale histograms.

**Figure 3 plants-14-01965-f003:**
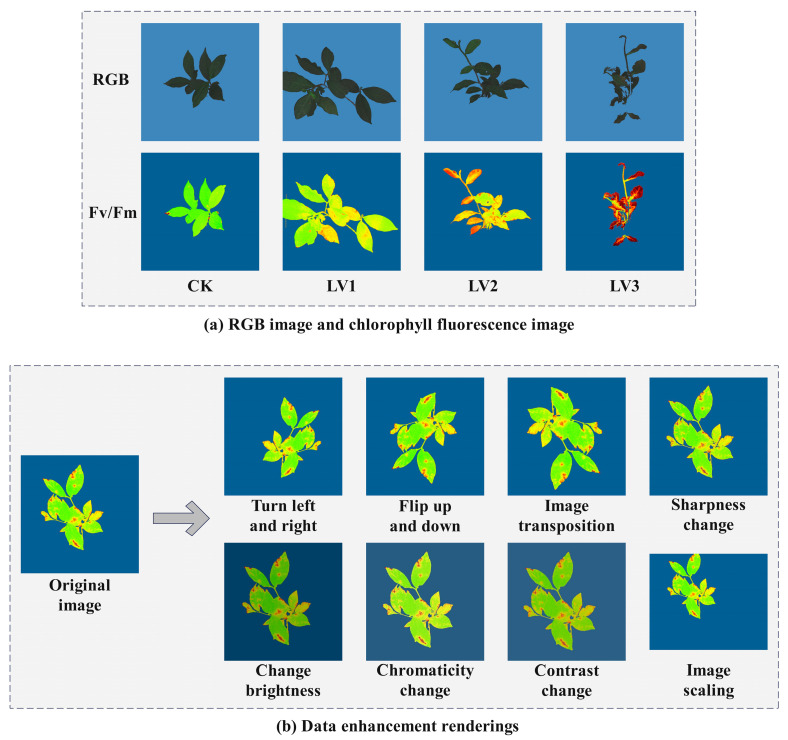
Display of image data samples.

**Figure 4 plants-14-01965-f004:**
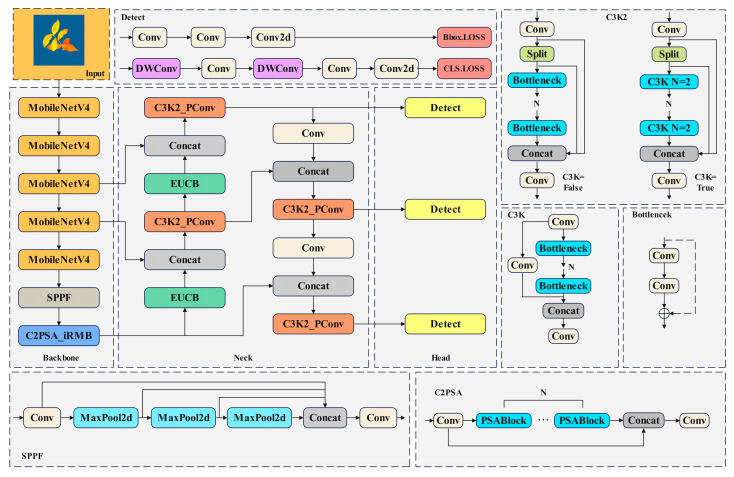
Improved YOLOv11 structure diagram.

**Figure 5 plants-14-01965-f005:**
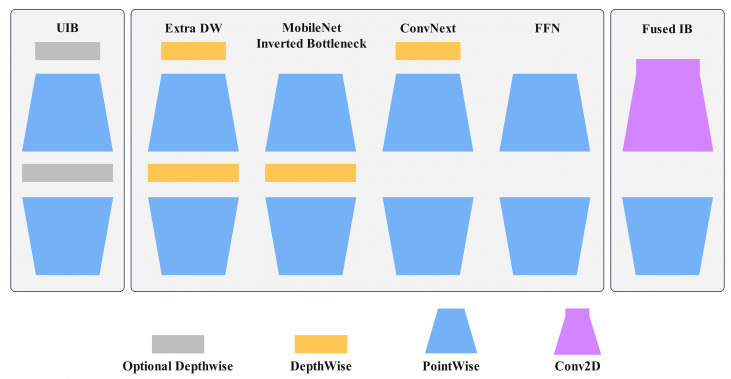
Universal Inverted Bottleneck blocks.

**Figure 6 plants-14-01965-f006:**
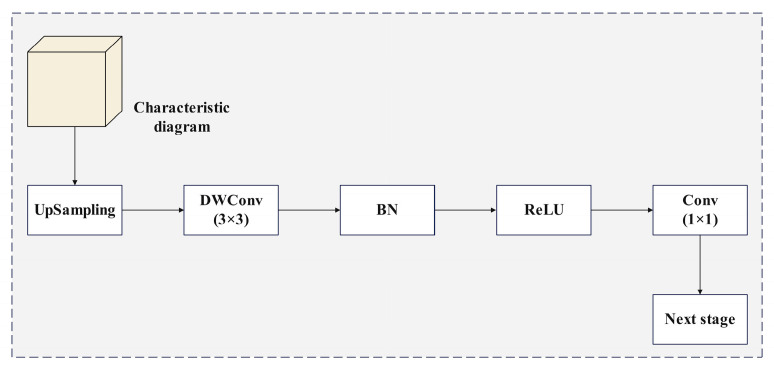
Efficient Up Convolution Block.

**Figure 7 plants-14-01965-f007:**
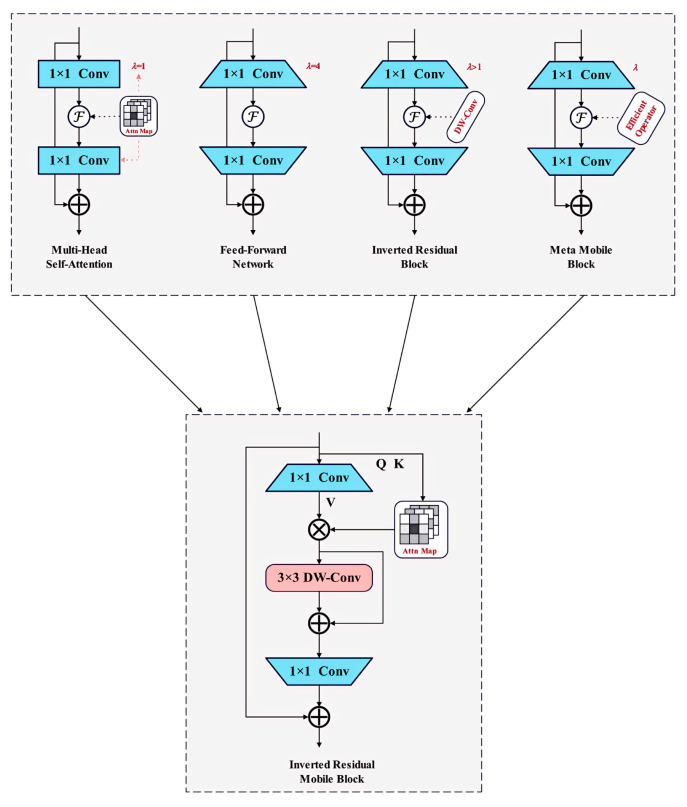
Inverted Residual Mobile Block.

**Figure 8 plants-14-01965-f008:**
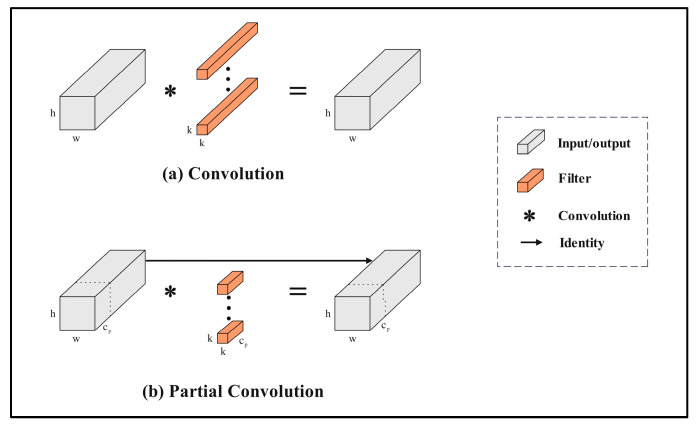
Partial Convolution.

**Figure 9 plants-14-01965-f009:**
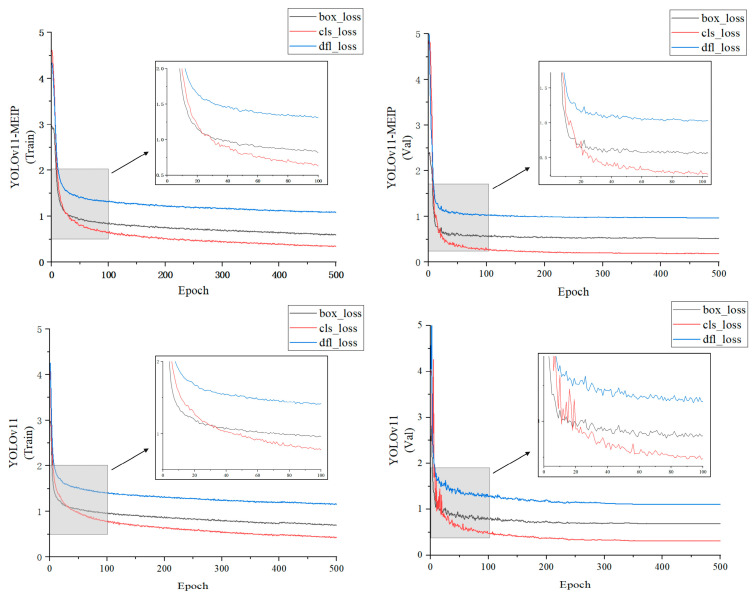
Comparison of loss function change curves.

**Figure 10 plants-14-01965-f010:**
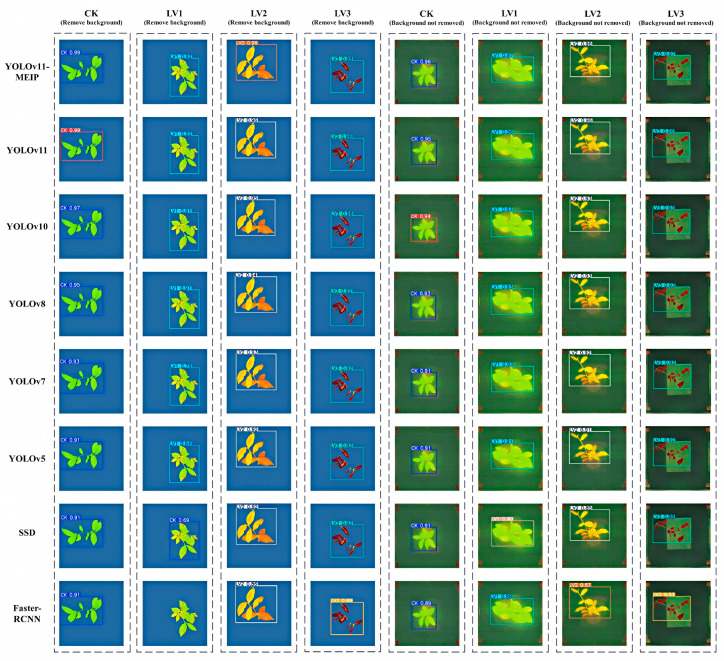
Comparison of detection results of different models.

**Table 1 plants-14-01965-t001:** Environmental settings of the temperature and humidity incubator.

Simulated Environment		Time/h	Environmental Air Temperature/°C			Relative Humidity/%
Healthy Culture	Daytime	16	25			80
	Nighttime	8	15			80
High-Temperature Stress	Daytime	16	30	36	42	80
	Nighttime	8	20	26	32	80

**Table 2 plants-14-01965-t002:** Chlorophyll fluorescence parameters and their meanings.

Fluorescence Parameter	Meaning	Fluorescence Parameter	Meaning
Fo	Basal fluorescence	qP	Photochemical quenching, established based on the “swamp model”
Fm	Maximum fluorescence	qN	Non-photochemical quenching coefficient
Fv/Fm	Maximum photochemical efficiency	qL	Photochemical quenching, established based on the “lake model”
Fm’	Maximum fluorescence under light adaptation	Y(NO)	Quantum yield of non-regulatory energy dissipation
Fq’/Fm’	Effective photochemical quantum yield of Photosystem II	Y(NPQ)	Quantum yield of regulatory energy dissipation
ETR	Electron transport rate	NPQ	Non-photochemical quenching coefficient
Fo’	Minimum fluorescence under light adaptation		

**Table 3 plants-14-01965-t003:** Correlation between chlorophyll fluorescence parameters and high-temperature stress levels. Note: an asterisk (*) indicates *p* < 0.05, double asterisks (**) indicate *p* < 0.01, and triple asterisks (***) indicate *p* < 0.001.

Fluorescence Parameter	Correlation Coefficient	Significance	Fluorescence Parameter	Correlation Coefficient	Significance
Fo	0.126		qP	−0.713	**
Fm	−0.486		qN	0.183	
Fv/Fm	−0.886	***	qL	−0.538	*
Fm’	−0.651	**	Y(NO)	0.843	***
Fq’/Fm’	−0.804	***	Y(NPQ)	0.307	
ETR	−0.804	***	NPQ	−0.054	
Fo’	0.032				

**Table 4 plants-14-01965-t004:** Operating system parameters.

Configuration Item	Configuration Parameter
Operating system	Windows 11
CPU	Intel(R) Core (TM) i7-14650HX
Memory	16 GB
GPU	NAIDIA GeForce RTX 4060
Compiled language	Python 3.8.19
Software framework	PyCharm 2024
CUDA	CUDA Version 11.8

**Table 5 plants-14-01965-t005:** Ablation study comparison results. Note: M represents the improvement of MobileNetV4; E represents the improvement of Efficient Up Convolution Block; I represents the improvement of Inverted Residual Mobile Block; and P represents the improvement of Partial Convolution.

Model	P/%	R/%	mAP50/%	Layers	Parameters	Gradients	GFLOPs	Weight/MB
YOLOv11	95.20	91.33	96.04	319	2593740	2593724	6.5	5.4
YOLOv11-M	96.10	90.77	97.21	386	2173383	2173367	5.3	3.8
YOLOv11-E	96.14	86.29	97.54	333	2680268	2680252	6.9	5.4
YOLOv11-I	97.14	97.90	98.91	333	2609612	2609596	6.5	5.2
YOLOv11-P	96.13	97.59	98.31	315	2510460	2510444	5.9	5.0
YOLOv11-ME	94.78	94.07	96.77	400	1886956	1886940	4.8	3.9
YOLOv11-MI	97.01	90.81	96.96	400	1816300	1816284	4.3	3.8
YOLOv11-MP	97.08	94.04	97.75	384	1727372	1727356	4.0	3.6
YOLOv11-EI	98.25	95.91	97.82	347	2696140	2696124	6.9	5.4
YOLOv11-EP	98.04	94.65	97.68	329	2596988	2596972	6.4	5.2
YOLOv11-IP	97.75	96.34	97.52	329	2526332	2526316	5.9	5.0
YOLOv11-MEI	98.39	97.45	98.79	414	1902828	1902812	4.8	3.9
YOLOv11-MEP	98.76	98.71	98.77	398	1813900	1813884	4.4	3.7
YOLOv11-MIP	98.73	98.38	98.74	398	1743244	1743228	4.0	3.6
YOLOv11-EIP	99.18	99.00	99.25	343	2612860	2612844	6.4	5.2
YOLOv11-MEIP	99.25	99.19	99.46	412	1829772	1829756	4.5	3.8

**Table 6 plants-14-01965-t006:** Comparative results of different models.

Model	P/%	R/%	mAP50/%	GFLOPs	Parameters	Weight/MB
YOLOv11-MEIP	99.25	99.19	99.46	4.5	1829772	3.8
YOLOv11	95.20	91.33	96.04	6.5	2593740	5.4
YOLOv10	94.49	91.48	97.2	8.2	2498168	5.1
YOLOv8	93.14	90.59	92.86	8.2	3011628	5.9
YOLOv7	91.36	88.29	91.17	105.2	37212738	71.3
YOLOv5	90.18	90.79	93.90	16.0	7030417	13.7
SSD	93.71	92.84	94.52	34.8	35641826	136.0
Faster-RCNN	92.77	93.62	95.23	134.38	41755286	159.7

## Data Availability

The data presented in this study are available upon request from the corresponding author.

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
