# Peer review of "Intelligent Identification of Tea Plant Seedlings Under High-Temperature Conditions via YOLOv11-MEIP Model Based on Chlorophyll Fluorescence Imaging"

_plants, 2025, doi:10.3390/plants14131965_

Round 1
Reviewer 1 Report
Comments and Suggestions for Authors
MAJOR REVISION
The study titled "Intelligent Recognition of a Lightweight YOLOv11-MEIP Model for Tea Plant Seedlings under High-Temperature Conditions in Chlorophyll Fluorescence Imaging" is suitable for this journal. However, there are several areas of confusion in the results, discussion and conclusions sections that require revision by the author. I recommend that this article undergo a Major revision before acceptance. The followings are my comments.
Section/ line no. (#) Major Comments
- I am strongly recommend to check the whole manuscript for language clarity and grammatical mistake with the help of a professional language editing service.
- The author must be revise the title in positive way and must be simplify it.
- In abstract line 17-18 rephrase it.
- Explain YOLOv11 model in detail.
- No need to use redundant words throughout the manuscript for example, our works, our study etc. please remove all the redundant words.
- Revise all the keywords.
- Scientific name should be in italic formats.
- Line 58-59 should be revise it. .
- The aims of the study is too difficult to understand. The authors should provide a justification for using these specific technology, supported by relevant references from previous studies that have employed.
- How the chlorophyll fluorescence images were acquired and also, specify whether the dataset is public or proprietary.
- Can the author verify the result with actual field or pot experiment result?
- Simplify the methodology section and remove unnecessary figures from this section. Add more figures in the result section, this will be strengthen and quality of the manuscript.
- The discussion could be deepened by comparing the results with similar studies and exploring the potential mechanisms through this technology.
- I don’t find any limitation of the study such as scalability to field conditions, variability in imaging environments.
- The conclusion should be precise, informative and avoid adding unnecessary sentences. Directly state the key findings of this study and emphasize the significant contributions it offers.
I am strongly recommend to check the whole manuscript for language clarity and grammatical mistake with the help of a professional language editing service.
Author Response
Thanks very much for your time to review this manuscript. We’re really appreciate your comments and suggestions. We have considered these comments carefully and tried our best to address every one of them.
- I am strongly recommend to check the whole manuscript for language clarity and grammatical mistake with the help of a professional language editing service.
Response:
We sincerely appreciate your valuable suggestions on the paper. We highly regard your advice regarding language editing and fully understand your expectation that our paper should be presented to readers in clearer and more precise language. Your feedback is of great importance to us.
Prior to this submission, all authors have conducted multiple rounds of meticulous content and language proofreading of the manuscript. We have made every effort to ensure grammatical accuracy and clarity of expression to meet the journal's language requirements. After carefully considering your suggestions, we commit to conducting another thorough internal language review alongside other content revisions. We will pay special attention to grammatical accuracy, sentence structure, and overall fluency, and will invite colleagues proficient in English to assist in a detailed recheck.
We believe that this additional, meticulous internal review will fully ensure the language quality of the manuscript. We hope the revised version will meet both your and the journal's language standards.
- The author must be revise the title in positive way and must be simplify it.
Response:
Thank you very much for your valuable suggestions on our paper. We have made proactive and simplified modifications to the title according to your advice. The revised English title is “Intelligent Identification of Tea Plant Seedlings Under High-Temperature Conditions via YOLOv11-MEIP Model Based on Chlorophyll Fluorescence Imaging.”
We have eliminated the lengthy and redundant descriptors in the original title, making it more succinct and to-the-point while highlighting the core content and focus of our research. This is the intelligent identification of tea plant seedlings under high - temperature conditions using the YOLOv11 - MEIP model based on chlorophyll fluorescence imaging. Meanwhile, we have retained the key information to ensure that readers can quickly grasp the main idea of the article.
- In abstract line 17-18 rephrase it.
Response:
Thank you very much for your valuable comments on our manuscript. Based on your suggestions, we have revised lines 17-18 of the abstract. The revised text is as follows: “To achieve efficient, non-destructive, and intelligent identification of tea plant seedlings under high-temperature stress, this study proposes an improved YOLOv11 model based on chlorophyll fluorescence imaging technology for intelligent identification.” We have restructured the sentences to make them more concise and clear. Additionally, we have optimized some of the wording to enhance the fluency and accuracy of the language.
- Explain YOLOv11 model in detail.
Response:
We truly value your insightful comments on our paper. Regarding your suggestion to detail the YOLOv11 model explanation, we've given it due consideration. Our paper already covers the model's essential aspects, such as its background, major improvements, module functions, and innovative design. This should give readers enough background knowledge and strongly support our experimental design and conclusions. While a deeper dive into the model's principles and details could add depth and comprehensiveness, we're somewhat constrained by the paper's scope and focus. Therefore, we've decided to concentrate on the model's practical application and effectiveness in this study. This way, readers can better grasp its advantages and value in addressing the issue of identifying heat-stressed tea plant seedlings.
- No need to use redundant words throughout the manuscript for example, our works, our study etc. please remove all the redundant words.
Response:
Thank you for your valuable feedback on our manuscript. We have thoroughly reviewed and removed all redundant words. The entire manuscript has been meticulously revised to ensure all repetitions are eliminated. We understand that concise language is crucial for enhancing readability and professionalism, allowing readers to better comprehend the research.
- Revise all the keywords.
Response:
Thank you for your valuable feedback on the manuscript. We have modified all keywords as per your suggestion. The revised keywords are: Chlorophyll Fluorescence Imaging; High-Temperature Stress; Tea Plant Seedlings; Improved YOLOv11; Model Lightweight; Image Classification Recognition. These changes ensure the keywords better capture the paper's essence and research priorities.
- Scientific name should be in italic formats.
Response:
Thanks for your comments on the manuscript. As per your suggestion, we conducted a comprehensive review and modification of the Scientific name in the manuscript, changing them to italics.
- Line 58-59 should be revise it.
Response:
Thank you for pointing out the need for revision in lines 58-59. We have modified the sentence from “The tea-growing regions in China are mainly distributed in subtropical and tropical areas.” to “The tea-growing regions in China are primarily located in subtropical and tropical zones.”
- The aims of the study is too difficult to understand. The authors should provide a justification for using these specific technology, supported by relevant references from previous studies that have employed.
Response:
Thank you very much for your valuable comments. We have revised and summarized the research objectives to make them clearer and more explicit. Additionally, the rationale for using these specific techniques is mentioned in the Introduction section of the paper, and we have also included references to some relevant previous studies.
- How the chlorophyll fluorescence images were acquired and also, specify whether the dataset is public or proprietary.
Response:
First and foremost, we would like to extend our sincere gratitude for the valuable comments on our manuscript. The issues you raised are highly critical and of significant importance to enhancing the quality of our paper.
Regarding the acquisition of chlorophyll fluorescence images, in this study, we employed the PlantExplorer series multi-functional plant photosynthetic phenotyping measurement system. This system is equipped with innovative multi-spectral chlorophyll fluorescence and visible light imaging technologies, capable of simultaneously acquiring RGB, chlorophyll, anthocyanin, and chlorophyll fluorescence imaging, with an imaging area of 40 cm × 53 cm. To ensure that the reaction centers of Photosystem II (PSII) are fully open, a dark adaptation process of approximately 20 minutes is conducted on tea plant seedlings before the acquisition of chlorophyll fluorescence data. Detailed information has been elaborated in sections "2.2 Fluorescence Image Acquisition Equipment" and "2.3 Data Acquisition and Processing" of the manuscript.
As for the nature of the dataset, the dataset used in this study was specifically collected and created by our research team as a proprietary dataset, rather than a public one. In our experiments, we set four temperature levels at 25℃, 30℃, 36℃, and 42℃, and acquired chlorophyll fluorescence images of tea plant seedlings under different high-temperature stress conditions using the multi-functional plant photosynthetic phenotyping measurement system, resulting in a total of 244 raw images. To expand the scale of the dataset, reduce the risk of overfitting during the model training phase, and enhance the model's generalization ability, we performed a series of image augmentation techniques on the original data, including horizontal flipping, rotation transformation, brightness adjustment, and changes in chroma, contrast, and sharpness. The expanded dataset comprises 1,952 images, which were randomly divided into training, validation, and testing sets in the ratio of 8:1:1. The detailed process of dataset establishment has been described in section "2.3.3 Establishment of the Image Dataset" of the manuscript. Since our primary focus is on the specific physiological responses of Yunnan large-leaf tea seedlings ("Yunkang No. 10") under different high-temperature stress conditions, as well as the changes in chlorophyll fluorescence image characteristics, we have not yet found a public dataset that perfectly meets the needs of our study. Therefore, we chose to collect and construct our own proprietary dataset to ensure the high relevance and accuracy of the data to our research objectives.
- Can the author verify the result with actual field or pot experiment result?
Response:
Thank you very much for your valuable comments on our manuscript. We have given serious thought and analysis to the issues you raised, which are highly instructive. Due to limitations in experimental conditions and the research timeline, we were unable to conduct actual field or potted experiments to further validate the model's identification effects in this study. However, we fully recognize the importance and necessity of field or potted experiments in verifying laboratory research findings.
In our experimental design, we simulated high-temperature stress environments by setting different temperature levels in a temperature and humidity-controlled incubator, and subjected two-year-old Yunnan large-leaf tea seedlings ("Yunkang No. 10") to high-temperature stress treatments. We acquired chlorophyll fluorescence images of tea seedlings under varying degrees of high-temperature stress using chlorophyll fluorescence imaging technology and systematically studied and analyzed the dataset with the improved YOLOv11-MEIP model. The experimental results showed that the model achieved high precision and efficiency in identifying tea seedlings under different high-temperature stress levels, with Precision, Recall, and mAP50 reaching 99.25%, 99.19%, and 99.46%, respectively. Nevertheless, we are also aware of the differences between laboratory and actual field or potted environments, which may have some impact on the model's identification effects.
In the future, we plan to expand the scope of our research by conducting high-temperature stress experiments on tea seedlings under actual field or potted conditions to verify and optimize our model. By collecting chlorophyll fluorescence image data of tea plants under natural high-temperature conditions in the field or pots, we will further train and test the model to enhance its applicability and reliability in real agricultural production. This will help translate our research findings into better applications for tea plant high-temperature stress monitoring and control practices, providing tea farmers and agricultural managers with more accurate and effective technical support.
- Simplify the methodology section and remove unnecessary figures from this section. Add more figures in the result section, this will be strengthen and quality of the manuscript.
Response:
We sincerely appreciate your valuable comments and suggestions on our manuscript. We have carefully analyzed and discussed your review comments, and the following is our response to your suggested revisions。
Your suggestion regarding the simplification of the Methods section and the removal of unnecessary numbers is highly pertinent. Indeed, the Methods section contains some overly lengthy and cumbersome descriptions, and certain numerical details contribute little to the core content of the study, potentially creating a comprehension burden for readers. Therefore, we plan to streamline the Methods section by eliminating non-essential specific numbers and complex descriptions, focusing instead on highlighting the key points and implementation process of the research methods to make this section more concise, focused, and clear.
We understand your intention to enhance the quality of the paper by adding more figures, as they can indeed more intuitively present research results and aid readers in better understanding the findings. However, after consideration, we have decided not to add more figures in the Results section for the following reasons: (1) The current figures in the Results section already comprehensively display the key outcomes of the study, including comparisons of model performance, loss function change curves, and comparisons of detection effects among different models. We believe these figures are sufficient to clearly present the main findings and conclusions of the research, and additional figures might lead to redundancy and clutter. (2) The overall length of the paper needs to be controlled to ensure conciseness and readability. Given that we are already planning to simplify the Methods section, adding too many figures in the Results section might cause the paper to exceed a reasonable length, affecting the balance of the overall structure and the reader's experience.
Nevertheless, we highly value your suggestion and will certainly take into account the reasonable addition of figures in the Results section to enhance paper quality in our future manuscript writing. Meanwhile, we are open to your specific evaluations and further suggestions for improving the presentation of the current figures in the Results section to better perfect our paper.
- The discussion could be deepened by comparing the results with similar studies and exploring the potential mechanisms through this technology.
Response:
Thank you very much for your valuable suggestions. We have revised and supplemented the Discussion and Conclusions sections based on your advice, further highlighting the innovations and strengths of our study through comparisons with similar research. Additionally, we have delved deeper into the application of this technology in uncovering the underlying mechanisms of heat stress in tea plants, thereby enhancing the depth and breadth of the discussion.
- I don’t find any limitation of the study such as scalability to field conditions, variability in imaging environments.
Response:
Thank you for your high recognition of this study and your valuable suggestions. We are aware of the importance of pointing out the limitations of the study for the completeness of the paper. We have made corresponding modifications in the Discussion section, adding a discussion on the potential limitations of this study in practical applications, including the scalability under field conditions and the variability of the imaging environment. We hope that these modifications will better perfect the content of the paper and make the research more in-depth and comprehensive.
- The conclusion should be precise, informative and avoid adding unnecessary sentences. Directly state the key findings of this study and emphasize the significant contributions it offers.
Response:
Thank you very much for your valuable suggestions. We have re-examined the Conclusions section and streamlined and optimized it, removing redundant content and focusing directly on the key findings of this study. We have endeavored to make the conclusions more precise, informative, and highlight their significant contributions.

Reviewer 2 Report
Comments and Suggestions for Authors
The manuscript by Wang et al. is devoted to development of methods of revealing influence of high temperature on tea plant seedling through using chlorophyll fluorescence imaging. Potentially, this work seems to be interesting; however, I have comments and question.
Main point:
- The work seems to be confused. Authors used the sophisticated PAM method (on basis of PlantExplorer) to form Fv/Fm images. Using the PAM method in field conditions is limited (e.g., only small distances can be used for these measurements because saturation pulses should be provided). It means that Fv/Fm images are not “raw fluorescence images”. It is not clear: Can results of analysis of Fv/Fm images by the Lightweight YOLOv11-MEIP Model be used in real tea cultivation? Moreover, it is not clear: Can the Lightweight YOLOv11-MEIP Model be used to estimate level of heat stress? Finally, why was current florescence not used to estimate plant heat stress? This current fluorescence is similar to sunlight-induced fluorescence; i.e. it is more perspective for real plant cultivation.
Specific points:
- Introduction: The “optimized lightweight YOLOv11 model” should be described in more detail.
- Introduction: Research contributions (maybe, highlights) of this manuscript (P. 4, lines 146-158) seem to be confused; especially (1) and (2). For example, sentence “breaking through the previous research paradigm limited to the measurement of basic photosynthetic indicators” is not clear because Fv/Fm is typical and well-known “basic photosynthetic indicator”, which are used in numerous works. Sentence “This approach offers a novel perspective for understanding the mechanisms of heat-induced damage in tea plants” is not also clear. Analysis of mechanisms of heat-induced damage in tea plants seems to be absent in the work. Many other sentences in this part seem to be also confused. It should be checked and corrected.
- Section 2.1: Parameters of illumination (light spectra or type of light (white, red, blue, etc.), light intensity in μmol m-2 s-1) should be described in detail.
- Table 1: Four variants of “High-Temperature Stress” (not three) are shown in this table. At that, variant 25/15, which is fully corresponded to variant 25/15 in “Healthy Culture”, seems to be error. It should be checked.
- Section 2.2: Fluorescence measurements should be described in detail. What was type of actinic light used? What was intensity of this actinic light? What duration of this actinic light before measurement of Fq’/Fm, ETR, qN, qP, qL, etc? These intensity and duration can influence dynamics of photosynthetic parameters after dark adaptation. As a result, non-stationary changes in these parameters can be reasons of their low correlation to heat stress levels. Type of light (white, blue, red, etc) can influence to light absorbance by leaflets; i.e. it can influence to ETR (if ETR was calculated on basis of the standard equation). What were equations used to calculate photosynthetic parameters? They seem to be absent in the manuscript.
- Table 3 and Figure 2: Calculation of correlation coefficients requires quantitative values of “high-temperature stress levels”. What were quantitative values used? It should be described in detail. Now, this point is not clear.
- Table 3 and Figure 2: Table and Figure show same values. I suppose that the Table or the Figure should be eliminated.
- Table 3 and Figure 2: Significant correlation coefficients should be shown.
- Table 3 and Figure 2: Why did heating weakly influence qN/NPQ? NPQ is often sensitive to action of stressors.
Author Response
Thanks very much for your time to review this manuscript. We’re really appreciate your comments and suggestions. We have considered these comments carefully and tried our best to address every one of them.
- The work seems to be confused. Authors used the sophisticated PAM method (on basis of PlantExplorer) to form Fv/Fm images. Using the PAM method in field conditions is limited (e.g., only small distances can be used for these measurements because saturation pulses should be provided). It means that Fv/Fm images are not “raw fluorescence images”. It is not clear: Can results of analysis of Fv/Fm images by the Lightweight YOLOv11-MEIP Model be used in real tea cultivation? Moreover, it is not clear: Can the Lightweight YOLOv11-MEIP Model be used to estimate level of heat stress? Finally, why was current florescence not used to estimate plant heat stress? This current fluorescence is similar to sunlight-induced fluorescence; i.e. it is more perspective for real plant cultivation.
Response:
Thank you very much for your review of this study and your valuable suggestions. We have had detailed discussions and analyses regarding the issues you raised, and here are our responses.
(1) You pointed out that the manuscript seems rather confusing. We believe this might be due to unclear explanations of the research methods and procedures. In fact, this study is based on chlorophyll fluorescence imaging technology to obtain fluorescence images and related parameters of tea seedlings under different high-temperature stresses. Through Spearman rank correlation analysis, we identified the parameter Fv/Fm, which has the highest correlation with the level of heat stress. We then used the corresponding fluorescence images of Fv/Fm as the input dataset for the deep learning model and constructed a lightweight YOLOv11-MEIP model to achieve intelligent identification of heat stress in tea seedlings. This research process is systematic and coherent, aiming to provide an efficient, precise, and intelligent method for monitoring heat stress in tea seedlings. In future revisions of the manuscript, we will provide clearer and more detailed explanations of the research methods and procedures to avoid any confusion for readers.
(2) You mentioned the use of the complex PAM method (based on PlantExplorer) to form Fv/Fm images and pointed out the limitations of using the PAM method under field conditions. We fully understand your concerns. Indeed, the PAM method, which requires the provision of saturating pulses, is typically limited to measurements over relatively short distances and thus may not be directly applicable to large-scale tea plantation scenarios. However, in this study, we used the PlantExplorer series multi-functional plant photosynthetic phenotyping measurement system to obtain Fv/Fm images mainly for the precision and controllability of experimental research. Under laboratory conditions, with precise control of environmental factors such as temperature, humidity, and light, we can more accurately simulate the effects of different levels of heat stress on tea seedlings, thereby providing a reliable data basis for studying the heat stress response mechanisms in tea seedlings.
(3) Regarding the potential application of the lightweight YOLOv11-MEIP model's analysis results of Fv/Fm images in actual tea cultivation, we believe there is considerable potential. Although the current study was conducted in a controlled laboratory environment, chlorophyll fluorescence imaging technology, as a non-destructive detection method, has significant advantages in monitoring plant physiological status. The Fv/Fm parameter, as an important indicator reflecting plant photosynthetic efficiency and photoinhibition status, can to some extent characterize the degree of heat stress experienced by tea seedlings. By inputting Fv/Fm images into the lightweight YOLOv11-MEIP model, we can achieve rapid and precise identification of heat stress in tea seedlings. In actual tea cultivation, if portable or drone-mounted chlorophyll fluorescence imaging devices could be combined, it would be possible to achieve real-time monitoring and assessment of heat stress in tea plants in the field. We will further explore how to apply this technology to actual tea cultivation scenarios and optimize the model to adapt to complex field conditions in future research.
(4) Concerning the lightweight YOLOv11-MEIP model's capability to estimate heat stress levels, in this study, we analyzed chlorophyll fluorescence images of tea seedlings under different levels of heat stress and achieved precise identification of heat stress levels using the lightweight YOLOv11-MEIP model. The experimental results showed that the model performed excellently in terms of Precision, Recall, and mAP50, indicating its ability to effectively recognize the fluorescence image features of tea seedlings under different heat stress levels. This suggests that the lightweight YOLOv11-MEIP model has the potential to estimate heat stress levels, as it can judge the heat stress status of tea seedlings by analyzing changes in the Fv/Fm parameter within the fluorescence images. However, we also recognize that further multi-species, multi-environmental experimental studies are needed to apply this model to actual heat stress level estimation, combined with more physiological indicators for comprehensive evaluation to enhance the model's applicability and reliability.
(5) You mentioned that current fluorescence was not used to estimate plant heat stress and pointed out that this type of fluorescence, similar to sun-induced fluorescence (SIF), is more conducive to real plant cultivation. The current fluorescence you referred to is commonly known as "sun-induced fluorescence" (SIF). Indeed, SIF, emitted by plants under natural light conditions, is closely related to the actual photosynthetic process and can reflect the plant's energy utilization and photoprotection status under natural conditions. It is significant for monitoring plant physiological status and environmental responses and is more in line with actual tea cultivation environments. We chose to use the PAM method based on PlantExplorer to form Fv/Fm images mainly to precisely control experimental conditions during the research phase and to deeply explore the impact mechanisms of heat stress on the photosynthetic light reactions in tea seedlings. The PAM method can accurately measure the maximum photochemical efficiency (Fv/Fm) and other chlorophyll fluorescence parameters under dark adaptation or known light conditions, which are widely used in plant physiological ecology research to assess plant photosynthetic performance and stress levels. By analyzing the changes in these parameters, we can gain a deeper understanding of the damage mechanisms of heat stress on the photosystems of tea seedlings and provide a reliable theoretical basis for subsequent model construction and identification algorithm development.
However, we recognize the potential and advantages of SIF in actual tea cultivation. In fact, SIF remote sensing technology has shown broad application prospects in large-scale vegetation productivity estimation, crop yield prediction, and plant stress monitoring in recent years. In future research on tea plant heat stress monitoring, we plan to combine SIF technology with chlorophyll fluorescence imaging technology to explore the use of SIF signals to estimate heat stress levels in tea plants. We will further optimize and expand our model to better adapt to the complex conditions in actual tea cultivation environments, providing tea growers with more practical and efficient monitoring tools.
- Introduction: The “optimized lightweight YOLOv11 model” should be described in more detail.
Response:
Thank you very much for your review of our manuscript and for raising the issue regarding the description of the “optimized lightweight YOLOv11 model” in the Introduction. We have emphasized the research background and significance in the Introduction section. However, we did not elaborate on the details of the model optimization and improvement in the Introduction, mainly to maintain a clear structure and focus, and to avoid distracting from the understanding of the research background and significance with too many technical details.
Nevertheless, we fully recognize the importance of a detailed description of the model optimization and improvement for understanding and evaluating our research. Therefore, in Section 2.4 of the Materials and Methods chapter, we have provided a comprehensive, in-depth, and detailed explanation of the optimized lightweight YOLOv11 model. This includes an introduction to the overall architecture of the optimized YOLOv11 model, the functions and roles of each improved module, and other aspects to ensure that readers can clearly understand the design principles and advantages of the model.
- Introduction: Research contributions (maybe, highlights) of this manuscript (P. 4, lines 146-158) seem to be confused; especially (1) and (2). For example, sentence “breaking through the previous research paradigm limited to the measurement of basic photosynthetic indicators” is not clear because Fv/Fm is typical and well-known “basic photosynthetic indicator”, which are used in numerous works. Sentence “This approach offers a novel perspective for understanding the mechanisms of heat-induced damage in tea plants” is not also clear. Analysis of mechanisms of heat-induced damage in tea plants seems to be absent in the work. Many other sentences in this part seem to be also confused. It should be checked and corrected.
Response:
Thank you very much for your valuable comments on the description of the research contributions in the Introduction section of our manuscript. Based on your suggestions, we have carefully reviewed and revised the research contributions, particularly merging and optimizing points (1) and (2) to ensure clearer and more accurate expressions, avoiding repetition or confusion, and focusing more on the actual work and achievements of this study.
The revised content is as follows: "We systematically analyzed the effects of heat stress on the photosynthetic light reactions of tea plants. We comprehensively evaluated 13 chlorophyll fluorescence parameters and through Spearman correlation analysis, the key indicator with the highest correlation to heat stress levels was determined as maximum photochemical efficiency (Fv/Fm). Furthermore, using chlorophyll fluorescence imaging analysis, we delved into the impact of heat stress on the chlorophyll fluorescence characteristics of tea plant seedlings."
- Section 2.1: Parameters of illumination (light spectra or type of light (white, red, blue, etc.), light intensity in ) should be described in detail.
Response:
Thank you very much for your valuable comments on the description of the illumination parameters in Section 2.1 of the manuscript. Based on your suggestions, we have provided a detailed supplement to the relevant content.
- Table 1: Four variants of “High-Temperature Stress” (not three) are shown in this table. At that, variant 25/15, which is fully corresponded to variant 25/15 in “Healthy Culture”, seems to be error. It should be checked.
Response:
Thank you for your review and correction of Table 1 in our manuscript. We have noted the issues you pointed out and have revised Table 1 accordingly in the paper.
- Section 2.2: Fluorescence measurements should be described in detail. What was type of actinic light used? What was intensity of this actinic light? What duration of this actinic light before measurement of Fq’/Fm, ETR, qN, qP, qL, etc? These intensity and duration can influence dynamics of photosynthetic parameters after dark adaptation. As a result, non-stationary changes in these parameters can be reasons of their low correlation to heat stress levels. Type of light (white, blue, red, etc) can influence to light absorbance by leaflets; i.e. it can influence to ETR (if ETR was calculated on basis of the standard equation). What were equations used to calculate photosynthetic parameters? They seem to be absent in the manuscript.
Response:
Thank you very much for your review and valuable suggestions on the fluorescence measurement section. Based on your advice, we have provided detailed supplements and explanations for the relevant content.
The fluorescence measurement device used in this study is the PlantExplorer Pro+, which features a built-in four-channel multispectral LED system as its photosynthetic light source. This system includes two channels of white light, red light, blue light, and far-red light. The multispectral LED system allows for precise spectral control and can be used to study plant responses to different light wavelengths. At 60 cm from the light source, the photosynthetic light intensity ranges from 100 to 600 and is adjustable. Before measuring parameters such as Fq'/Fm, ETR, qN, qP, and qL, tea seedling leaves are subjected to a 20-minute dark adaptation treatment to ensure that the reaction centers of Photosystem II (PSII) are fully open. After dark adaptation, the tea seedling leaves are illuminated with the photosynthetic light from the device for parameter measurement, which takes approximately 11 to 12 minutes. The multispectral characteristics of the photosynthetic light (white, red, blue, and far-red light) allow for the study of the effects of different light wavelengths on plant photosynthesis. Different light wavelengths have varying impacts on plant light absorption and photosynthesis, which may affect the calculation of parameters such as ETR. We have also supplemented the relevant sections of the manuscript with the formulas for calculating the photosynthetic parameters used in this study.
- Table 3 and Figure 2: Calculation of correlation coefficients requires quantitative values of “high-temperature stress levels”. What were quantitative values used? It should be described in detail. Now, this point is not clear.
Response:
Thank you very much for your valuable comments on the calculation of correlation coefficients in Table 3 and Figure 2. Based on your suggestions, we have provided a detailed supplement to the relevant content.
When calculating the Spearman rank correlation coefficient, the quantitative values of heat stress levels were based on the different heat stress levels set in the experimental design. In the correlation analysis, these heat stress levels were encoded as numerical values: 0 (CK), 1 (LV1), 2 (LV2), and 3 (LV3). By doing so, we were able to conduct a correlation analysis between heat stress levels and chlorophyll fluorescence parameters to identify which parameters had significant correlations with heat stress levels.We have supplemented the manuscript with this information to ensure that readers can clearly understand the basis for the calculation of the correlation coefficients.
- Table 3 and Figure 2: Table and Figure show same values. I suppose that the Table or the Figure should be eliminated.
Response:
Thank you very much for your review and valuable comments on Table 3 and Figure 2. Based on your suggestions, we have removed Figure 2 to avoid redundant presentation of the same information.
- Table 3 and Figure 2: Significant correlation coefficients should be shown.
Response:
Thank you very much for your review and valuable comments on Table 3 and Figure 2. Based on your suggestions, we have removed Figure 2 and added significance annotations in Table 3 to highlight the significant correlation coefficients.
- Table 3 and Figure 2: Why did heating weakly influence qN/NPQ? NPQ is often sensitive to action of stressors.
Response:
Thank you very much for your review and valuable suggestions. Regarding the issue that the effect of high-temperature stress on qN/NPQ is relatively weak, we have conducted an in-depth analysis and referred to relevant literature. Here is our response.
NPQ (Non-photochemical quenching) is generally sensitive to stress factors. However, in this study, the effect of high-temperature stress on qN/NPQ was relatively weak, which we analyzed to be related to the following reasons: (1) The steady-state value of NPQ is partly dependent on temperature. However, when the temperature exceeds a specific threshold, the capacity of NPQ can become saturated, which may limit its response to high-temperature stress. In this study, the range of high-temperature stress set may have exceeded the sensitive response interval of NPQ, resulting in insignificant changes in NPQ. (2) Complex Effects of High Temperature on Photoprotection Mechanisms. NPQ is one of the important photoprotective mechanisms in plants. However, under high-temperature conditions, plants may activate multiple photoprotective mechanisms to cope with photoinhibition. These complex photoprotective mechanisms may, to some extent, mask the response of NPQ to high-temperature stress. (3) The statistical characteristics of the data may affect the results of correlation analysis. Variability among biological individuals may also lead to inconsistent changes in NPQ. Different tea seedling individuals may have different responses to high-temperature stress, which may also result in inconsistent changes in NPQ.

Round 2
Reviewer 1 Report
Comments and Suggestions for Authors
I have carefully reviewed the revised manuscript, which has shown some improvement. However, there are still minor suggestions that the authors must address before publication. Furthermore, the point-by-point responses to the comments are not properly structured, the authors should include line numbers indicating where revisions were made (e.g., from line X to line Y). Additionally, unnecessary sentences throughout the manuscript should be removed. Below are some specific concerns that require revision.
- I still don’t find any information about the YOLOv11 model in in introduction part.
- I strongly recommended to verify the result of chlorophyll content analysis from pot experiment because no one can agree the result of the machine without the verification of a real life.
- Still the aims and objective are not clear, the author must be revise with clear hypothesis.
- The discussion need to be revise and correlate your result with previous published study. No need to be repeated the result.
Author Response
1、I still don’t find any information about the YOLOv11 model in in introduction part.
Response:
Thank you for your valuable comments on our paper. You pointed out that there was no information about the YOLOv11 model in the introduction section, which we have carefully considered and revised.
We agree that a brief introduction of the YOLOv11 model in the Introduction section is necessary to help readers better understand the background and methods of our study. Therefore, we have added a short introduction to the YOLOv11 model in the Introduction section to provide sufficient background information while avoiding excessive text. The revised content of the Introduction section is as follows (lines 129 to 142 in the manuscript): Concurrently, with the rapid advancement of deep learning technology, object detection algorithms have been widely applied across multiple fields. The YOLO series of algorithms, as classic models in the realm of real-time object detection, have garnered extensive attention due to their swift detection speeds and high levels of detection accuracy. YOLOv11, the latest iteration of this series, has achieved a significant enhancement in both precision and speed compared to its predecessors. It employs an improved version of CSPDarknet53 as its backbone network and introduces the Spatial Pyramid Pooling Fast (SPFF) module and the C2PSA module with pyramid slicing attention mechanism, thereby further bolstering its feature extraction capabilities. Additionally, the detection head of YOLOv11 adopts a decoupled structure, selecting appropriate loss functions based on different tasks, which markedly improves the model's performance. These refinements enable YOLOv11 to excel in processing large-scale image data, thereby furnishing a robust technological foundation for our research.
By adding this brief introduction in the Introduction section, we aim to provide readers with sufficient background information without taking up too much space, so that they can better understand the reasons and significance for choosing the YOLOv11 model for identifying the heat stress status of tea seedlings. Thank you again for your valuable comments. We believe that these revisions will make our paper more complete and clear.
2、I strongly recommended to verify the result of chlorophyll content analysis from pot experiment because no one can agree the result of the machine without the verification of a real life.
Response:
Thank you very much for reviewing our paper and for your valuable comments. Your suggestion regarding the validation of chlorophyll content analysis results from potted plant experiments is extremely important, and we fully understand your concern about the real-world validation of model results.
In our study, our objective is to achieve efficient, non-destructive, and intelligent identification of the status of tea seedlings under different high-temperature stresses by combining chlorophyll fluorescence imaging technology with the improved YOLOv11-MEIP model. Our aim is to utilize chlorophyll fluorescence imaging, a non-destructive detection method, in conjunction with deep learning technology, to provide a rapid and accurate monitoring method for the high-temperature stress status of tea seedlings. This approach not only minimizes physical damage to plants but also enables real-time monitoring of physiological state changes in plants.
Although we did not conduct chlorophyll content analysis, our research methods and results still hold significant scientific and practical value. Chlorophyll fluorescence imaging, as an emerging non-destructive detection method, has shown great potential in monitoring plant physiological states. By examining changes in fluorescence parameters and images, we can indirectly reflect the physiological responses of plants under different environmental stresses. Our model, through the analysis of fluorescence images, can accurately identify the status of tea seedlings under various high-temperature stresses, offering new theoretical support and technical references for the monitoring and control of tea plants and other crops under high-temperature conditions.
To further enhance the persuasiveness of our study, we plan to combine chlorophyll content analysis with the validation of model prediction results in future research. This will help us gain a deeper understanding of the relationship between chlorophyll fluorescence parameters and actual chlorophyll content, thereby further optimizing our model and methods. We will meticulously document and analyze these data in our subsequent research and report them in future papers.
Thank you once again for your valuable comments. We believe that they will help us to further refine our research and provide more comprehensive and accurate technical support for the field of tea plant high-temperature stress monitoring.
3、Still the aims and objective are not clear, the author must be revise with clear hypothesis.
Response:
Thank you very much for reviewing our paper and for your valuable comments. Following your suggestions, we have further clarified and refined the aims and objectives of our study to ensure that our research direction and expected outcomes are more explicit. In the abstract section (lines 16 to 18 in the manuscript), we have outlined the purpose of our research. We have also revised the aims and objectives in the Introduction section of the paper. The revised content is as follows (lines 146 to 154 in the manuscript): This study takes the "Yunkang 10" tea seedlings as the research object. By analyzing the fluorescence parameters and images of tea seedling leaves under different high-temperature stresses obtained through chlorophyll fluorescence imaging technology, and in combination with the proposed improved YOLOv11 model, it achieves efficient, non-destructive, and intelligent identification of the status of tea seedlings under various high-temperature stresses. This study aims to explore the potential application of chlorophyll fluorescence imaging and deep learning technology in the stress monitoring of tea seedlings, providing a rapid and accurate monitoring method for the high-temperature stress status of tea seedlings.
Through the revision of the aims and objectives, we hope to enable readers to have a clearer understanding of our research direction.
4、The discussion need to be revise and correlate your result with previous published study. No need to be repeated the result.
Response:
Thank you for your valuable comments on our paper. You pointed out that there was no information about the YOLOv11 model in the introduction section, which we have carefully considered and revised.
We agree that a brief introduction of the YOLOv11 model in the Introduction section is necessary to help readers better understand the background and methods of our study. Therefore, we have added a short introduction to the YOLOv11 model in the Introduction section to provide sufficient background information while avoiding excessive text. The revised content of the Introduction section is as follows (lines 129 to 142 in the manuscript): Concurrently, with the rapid advancement of deep learning technology, object detection algorithms have been widely applied across multiple fields. The YOLO series of algorithms, as classic models in the realm of real-time object detection, have garnered extensive attention due to their swift detection speeds and high levels of detection accuracy. YOLOv11, the latest iteration of this series, has achieved a significant enhancement in both precision and speed compared to its predecessors. It employs an improved version of CSPDarknet53 as its backbone network and introduces the Spatial Pyramid Pooling Fast (SPFF) module and the C2PSA module with pyramid slicing attention mechanism, thereby further bolstering its feature extraction capabilities. Additionally, the detection head of YOLOv11 adopts a decoupled structure, selecting appropriate loss functions based on different tasks, which markedly improves the model's performance. These refinements enable YOLOv11 to excel in processing large-scale image data, thereby furnishing a robust technological foundation for our research.
By adding this brief introduction in the Introduction section, we aim to provide readers with sufficient background information without taking up too much space, so that they can better understand the reasons and significance for choosing the YOLOv11 model for identifying the heat stress status of tea seedlings. Thank you again for your valuable comments. We believe that these revisions will make our paper more complete and clear.

Reviewer 2 Report
Comments and Suggestions for Authors
Authors considered my comments. I have not additional remarks.
Author Response
Authors considered my comments. I have not additional remarks.
Response:
We are truly grateful for your review of our paper and the valuable comments you have provided. We have carefully considered each of your suggestions and have made corresponding revisions and improvements in the paper. Your feedback has played a crucial role in enhancing the quality of our manuscript.
We are very relieved to see that you believe we have fully taken your comments into account. If you have any further suggestions or ideas in the future, please feel free to put them forward. We will continue to refine and optimize the paper in order to achieve a higher academic standard.
Thank you once again for taking the precious time to review our paper!
